# The Formation of the Urban–Rural Fringe Space in the San Cayetano Area: The Transformation of a Peripheral Urban Landscape in Ecuador

Karina Monteros [1], Valentina Dall'Orto [1,2] and Cecilia Cempini [1,2,*]

1   Department of Architecture and Urbanism, UTPL, Universidad Técnica Particular de Loja, San Cayetano Alto, Loja 110107, Ecuador; kmonteros@utpl.edu.ec (K.M.); vdallorto@utpl.edu.ec (V.D.)
2   Department of Architecture and Urban Studies, DAStU, Politecnico di Milano, 20133 Milan, Italy
*   Correspondence: ccempini@utpl.edu.ec

**Abstract:** The transition between the urban and the rural in intermediate Andean cities has been consolidated, presenting a close relationship between socioeconomic dynamics and urban configuration. The peripheral neighborhood of San Cayetano, located in the city of Loja in southwestern Ecuador, presents multiple contradictory scenarios. Located on a hillside, the forms of adaptation to the challenging topography are diverse, fragile, and fragmented, especially because they present a dispersed image due to the proximity to the consolidated center. This study analyzes the spatial phenomena that have led to the integration of this neighborhood into the formal city, identifying recurrent spatial configurations that characterize the spatial fragment as a determinant in the configuration of the Andean periphery. To this end, official data are juxtaposed with on-site visits to identify the urban and architectural patterns of the neighborhood that contribute to defining this characteristic fragmentation of today's peripheries. These patterns are then examined through mapping and graphic representation. As a consequence, the resulting urban plots are imposed on sloping land, leaving aside the natural characteristics of the terrain, which causes morphological alterations at the level of the natural, urban, and architectural landscape.

**Keywords:** rural–urban frontier; Andean city; spatial fragmentation





## 1. Introduction

Peri-urbanization is an emerging process within contemporary urban phenomena, impacting cities of varying sizes and geographies. The diffuse and dispersed forms of urbanization challenge the traditional conceptual dualism between rural and urban areas, creating a transitional space between them. In this dynamic, the rural environment acquires a more urban character in physical, economic, and social terms [1]. This dividing line between the rural and the urban entails the attributes that one has concerning the other, the urban being allusive to densely populated areas [2] characterized by the presence of buildings, public services, advanced infrastructures, and high population density. In contrast, rural areas are less developed, presenting low population density, predominantly agricultural activities, and less infrastructure. The concept of rural areas also includes a group of settlements below a certain population [3]. Both concepts present a difference in scale. Given the changing definition of rural areas, planners have begun to delve deeper into the interface between urban and rural areas. There is a closer connection between the city and the countryside, referred to in the literature as the "rural–urban area" [4].

The study of urban peripheries has evolved since the 1970s, with recent decades witnessing increased analysis and constant conceptual redefinition [5,6]. These studies recognize various processes, from the incorporation of new urban areas away from existing centers to the "peripheralization" of inner districts due to changing economic and social conditions [6]. Initially, for Montoya Arenas, the interest of academia was in resolving

questions concerning general demographic [7] dynamics, including urbanization processes, internal migration, informal settlements, and urban poverty, often categorized under the umbrella of marginality.

Currently, due to the heterogeneity of scenarios within the periphery, research is approached from different angles, encompassing social and economic sciences, as well as environmental sciences [8]. Soja [9,10] refers to this transitional boundary as a "third space" that encompasses several dimensions in the creation of space and territory, interpreted from the perspective of how the physical dimensions of an entity intersect with its social and cultural context. This notion is particularly relevant in the contemporary spatial discourse, which recognizes border phenomena as significant spatial entities that extend beyond their geopolitical boundaries. This phenomenon is also referred to as "rural–urban areas", characterized as localities with a historical structure that is both consolidated and rooted. However, their connection to the city lacks comprehensive integration, resulting in socio-territorial marginalization despite the passing of time [11]. This separation is so-called fragmentation, as a border beyond those demarcated by the imaginary lines of the cartography [12]. Fragmentation shows urban–rural contrasts as an incomplete process that deserves reflection and analysis.

This peri-urban area is intimately associated with the transition from a dense urban structure to a rural one [13], which the intermediate Andean cities present, making its analysis particularly interesting, since there is a lack of in-depth studies on the subject, although they are significant contributors to contemporary urban growth.

This situation is prevalent worldwide, particularly in Ecuador and Latin America [14], where informal settlements often arise due to similar factors like socioeconomic inequality, lack of access to basic services such as potable water and sanitation, and inadequate housing policies. Although this problem may vary in form and scale in different parts of the world [14,15], they share similar problems and challenges faced by people living in these places. These challenges include a lack of legal and land tenure security, vulnerability to natural disasters, and social exclusion, among others.

In Ecuador, population growth projections through 2023 indicate that cities in three regions—the highlands, the coast, and the Amazon—are experiencing significant urbanization. Specifically, cities such as Esmeraldas, Loja, and Orellana are highlighted for their higher rates of urbanization [16]. The proliferation of peripheries is a consequence of the application of neoliberal policies of central governments, as evidenced by [16] (p. 15). "The State concentrates its investments in those urban functions necessary for the functioning of the dominant pole, that is, capital. While the housing, transportation, and social equipment needs of the population are left aside".

Ecuador exhibits three defining aspects of the urban process: the crisis of banana exploitation (1965), changes in the agrarian structure of the Sierra due to agrarian reform (1970) [17], and the beginning of import substitution processes. These factors accentuated the poverty of the country's inhabitants, whose housing options oscillated between the use of old buildings in urban centers and the formation of spontaneous settlements in the peripheries [18]. The agrarian reform led to the differentiation of the peasantry into nobles and landless peasants, causing social and economic tensions [3]. The right of expropriation was included, which was based on the concept of public utility, and the large landowners were required by law to grant small plots of land to their workers. They also highlight the poverty of the country's residents, whose housing options range from the use of old buildings in urban centers to the creation of spontaneous settlements on the outskirts of the city [19]. The collapse of the banana industry caused a massive exodus of workers from the countryside to the cities, generating rapid urbanization without adequate planning, overcrowding in slums, lack of basic services, and increased crime. Uncontrolled urban sprawl affected the population's quality of life. This dynamic was the result of a neoliberal model aimed at capital accumulation through the restructuring of the formal city, integrating economic activities related to the tertiary sector and residential spaces.

Import substitution in Ecuador brought about several changes at the urban level, including greater industrial development, an increase in the supply of employment, and most importantly, the growth of the urban middle class. Industrialization and national production led to greater urban economic growth, creating cities of opportunity for urban dwellers [20]. Although the informal city is legally recognized, efforts to provide comprehensive support for its inhabitants often fall short, resulting in the implementation of mere "survival strategies" without addressing the underlying structural complexities [21–23].

Until the 1990s, the most significant spatial, social, and economic transformations occurred mainly in metropolises such as Quito and Guayaquil. However, since the 2000s, increased migration to the United States and Europe spurred direct investments in medium-sized urban centers, along with a gradual migration from rural areas to cities, reinforcing the country's cantonal structure [24]. Migration was, therefore, a very significant sign of the welfare and progress of a nation, which sought to find better job opportunities far from its native place [25]. Migrants have affected the development of cities by fostering the formation of the "poverty belt", exacerbated by poor governance, conflict, and social pressure. In intermediate cities, internal migration serves as a spatial phenomenon driven by the desire to enhance economic opportunities, leading individuals to abandon primary agricultural and livestock activities in favor of settling near urban centers, where employment prospects like construction work are more promising. Migration dynamics are influenced by local and regional power relations, indirectly motivating population movements [26–28].

Consequently, migrants tend to consider peri-urban spaces as ideal areas for settlement, whether by formal or informal means. Although research has traditionally focused on the peripheries of large cities, it is crucial to understand the dynamics of intermediate cities, such as the case of Ecuador. Several studies in the country explore peripheral neighborhoods from different perspectives: one suggests densification as a strategy to address socio-spatial conflicts [24], while another examines spatial production practices by contrasting planned and self-built cities in Riobamba and Cuenca [29].

Although there are some common characteristics, there is a great difference between peri-urban areas in developing countries characterized by land and water pollution, poverty, and informal settlements; and those in developed nations characterized by low levels of mobility, economic performance, landscape integrity, and environmental quality.

In South America, peripheral neighborhoods emerged in the 1970s [5,30] due to land subdivision by urban landowners, who took advantage of the growth conditions of a "demand" coming from the countryside or the renewal areas of the city, and of the urbanization of land promoted by the municipality. Population growth, migration, the progressive saturation of central areas, the high cost of land, and the inadequate production of social housing [30,31] are the main causes of the progressive fragmentation of rural areas around the city [32]. The common characteristic between these different scenarios is that people share a non-urbanized context, then, the forms of production of space that characterize it tend to be urbanity.

Given the multifaceted nature of urban peripheries, this study hypothesizes the existence of a spatial occupation that defies clear classification as rural or urban, representing a space characterized by uncertainty and incomplete understanding [33] termed the "blind field" by Lefebvre [34]. Focusing on San Cayetano, a peripheral neighborhood in Loja, Ecuador, this study analyzes the spatial phenomena that led to its integration into the formal city, identifying the stages of consolidation over time and their implications in terms of urban and architectural occupation. San Cayetano presents a heterogeneous layout, continuously evolving to adapt to different realities: spontaneous hillside settlements driven by urgent housing needs juxtaposed with more regular layouts reminiscent of the planned urban fabric, albeit adapted to the sloping terrain. Urban–rural fragmentation in hillside areas has led to a loss of connectivity, causing access and mobility problems for their inhabitants, as well as difficulties for biodiversity conservation and sustainable land management; while in Ecuador, the relationship between municipal administration and neighborhoods focuses on the provision of basic services, citizen participation, and

infrastructure improvement, although it does not always work in practice. Here, the non-existence of composition principles is clear, since the need for housing that satisfies the basic needs is urgent; on the other hand, a more regular layout that simulates the planned grid of the urban center, implanted but in a sloping topography in an indifferent way. To this end, we seek to understand the elements that characterize these processes of urban transformation in the Andean intermediate city, through direct observation of this case study, relating the monocentric neoliberal model with the socio-spatial phenomena and the diversity of ways of inhabiting that characterize it.

This research is pertinent as it seeks to incorporate traditionally excluded spaces into architectural discourse, recognizing peripheral spatiality as a dynamic, multiform phenomenon characterized by contradiction and coexistence. Understanding this environment necessitates the adoption of new analytical tools and the consideration of diverse spatial categories. San Cayetano exemplifies these contradictions, hosting both rural and urban activities blended in undefined natural and built barriers, formal and informal road networks, and architecture ranging from fragile and improvised to more formal, yet non-compliant with norms. By comprehending this process and identifying its developmental stages, it is possible to establish the urban-architectural foundations for the proposition of consolidative actions.

This article represents the initial output of an ongoing study, presenting questions and contradictions that will inform subsequent research phases. Economic aspects influencing the informal or formal appropriation of peripheries are excluded from this analysis, yet remain pertinent for future investigation.

## 2. Materials and Methods

The proposed descriptive and analytical methodology aims to identify urban-architectural constants in the emerging periphery of the intermediate Andean city and correlate these findings with variables dictated by the ongoing neoliberal model.

To achieve this goal, the process of transformation of the neighborhood from its inception to its current state is studied through the comparison of various sources, including cartographic data, demographic and socio-economic statistics, and empirical information such as social and living practices and space utilization. The characteristics of the periphery were categorized according to Allen's framework [35], which includes morphological and functional aspects, social and daily life dynamics, connectivity, and integration within an ecological mosaic.

The research, in its preliminary phase, covered six months of fieldwork and included various academic activities that facilitated direct contact with the territory and an understanding of its dynamics. The research process was structured in three consecutive phases, each of which contributed to a deeper understanding of the study area.

The first phase involved a literature review to identify formal and informal conditions arising in the peripheries as a consequence of the application of the neoliberal model in Ecuador. This included the characterization of peri-urban spaces in the intermediate cities, extending this classification to the realities of the neighborhood.

Data collection utilized a series of complementary approaches aimed at obtaining a comprehensive overview of the Loja peri-urban area. Initially, topographical and geographical maps sourced from local and national institutions were employed to gain a general understanding of the study area and to identify its key features, thereby comprehending the morphology of the territory and its influence on the built environment. The use of QGIS to analyze the occupation layers of urban and rural areas within the urban area of the site facilitated the identification of changes in occupation patterns over the years. The profile tool allowed the generation of the topographic contour, crucial for studying sloping hillsides and providing detailed information on the elevation of the terrain.

In addition, the superimposition of the different layers that make up the urban area made it possible to extract data regarding the distribution and concentration of built-up

areas in the study region. Subsequently, the final format of the map was created using graphic design programs such as Adobe Illustrator to ensure maximum expressiveness.

Socioeconomic data on income and labor situation at the parish level were included, sourced from previous research carried out by academic institutions, as well. Fieldwork allowed direct observation and documentation of the urban-architectural characteristics of the case studied. First, the analysis of the fragmented urban fabric of the site facilitated the identification of recurrent infrastructures, highlighting their adaptability to the complex topographic conditions of the area. In addition, various practices of domestic space appropriation were discerned by examining emblematic architectural examples found in the neighborhood. Essential operational tools, such as photographic documentation of the survey and on-site sketching, were employed to identify the distinctive features of the neighborhood. Photography was also used to assess urban sprawl and analyze the dynamics of landscape transformation.

The third phase focused on the systematization of the information collected and its analysis to identify patterns of spatial transformation. Re-elaboration and synthesis are carried out through the interpretation of maps that allow a comparison of their configuration process and a successive deduction of the spatial principles of the environment.

The superimposition of statistical data with in situ information has facilitated the hypothesis of correlations between urban phenomena and the socioeconomic dynamics that drive land use. The application of the methodology allowed a characterization of the spatial fragment in the study environment through a multiscale analysis, thus determining the distinctive features of the neighborhood unit and the elements of the public space that play a fundamental role in the consolidation process.

*Case Study: San Cayetano Neighborhood*

The city of Loja is located in the south of the Inter-Andean Region of the Republic of Ecuador, in the province of Loja, at an average altitude of 2065 m above sea level and 4 degrees south latitude (Figure 1). It has an area of 5732.51 hectares [36] and has 100,271 inhabitants according to the last population census of the National Institute of Statistics and Census (INEC) [37].

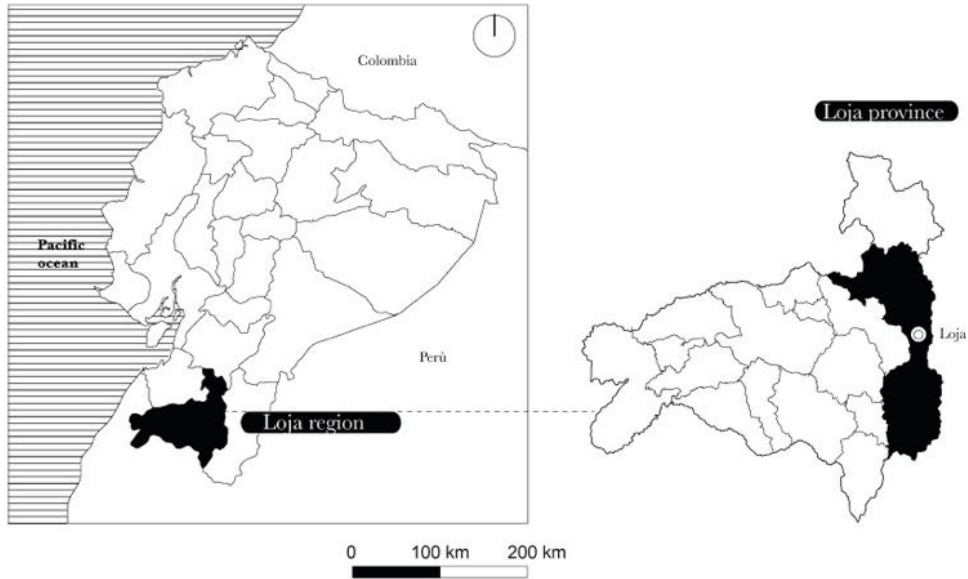

**Figure 1.** Loja is an intermediary city nestled in the southern Andean region of Ecuador. Serving as the capital of both the canton and the province bearing its name, Loja holds a pivotal position in the southern part of the country, playing a strategic role both regionally and nationally. Source: the authors (2024).

The San Cayetano neighborhood belongs to El Valle parish, located in the western part of the city. It has its origins in an Indigenous village settled in this area in 1624, initially administered by the Jesuit order as a religious doctrine called "San Juan del Valle" [38]. At this initial stage, the settlement took the urban pattern of the time—the square grid of growth based on the presence of the square and church. From its origin, this sector was located on the outskirts of the city center.

In terms of urban development, the city experienced a slow growth until 1960, with urban sprawl not extending beyond the natural boundaries of the Zamora and Malacatos rivers, encompassing roughly 30 to 40 blocks (Figure 2). However, by 1970, the city's footprint expanded to 556 hectares, marking a significant shift in its urban layout. This period signified a pivotal moment in the city's expansion with the implementation of the first Regulatory Plan, commissioned by the Uruguayan urban planner Gatto Sobral. This plan initiated a process of urban transformation by formally planning new urban developments beyond the traditional limits between the Zamora and Malacatos rivers, establishing connections between the established central area and the expanding peripheries [36]. This era of neoliberal urbanization emphasized the primacy of the economic sphere over natural resources, leading to a reduction in agricultural land on the city's periphery [39].

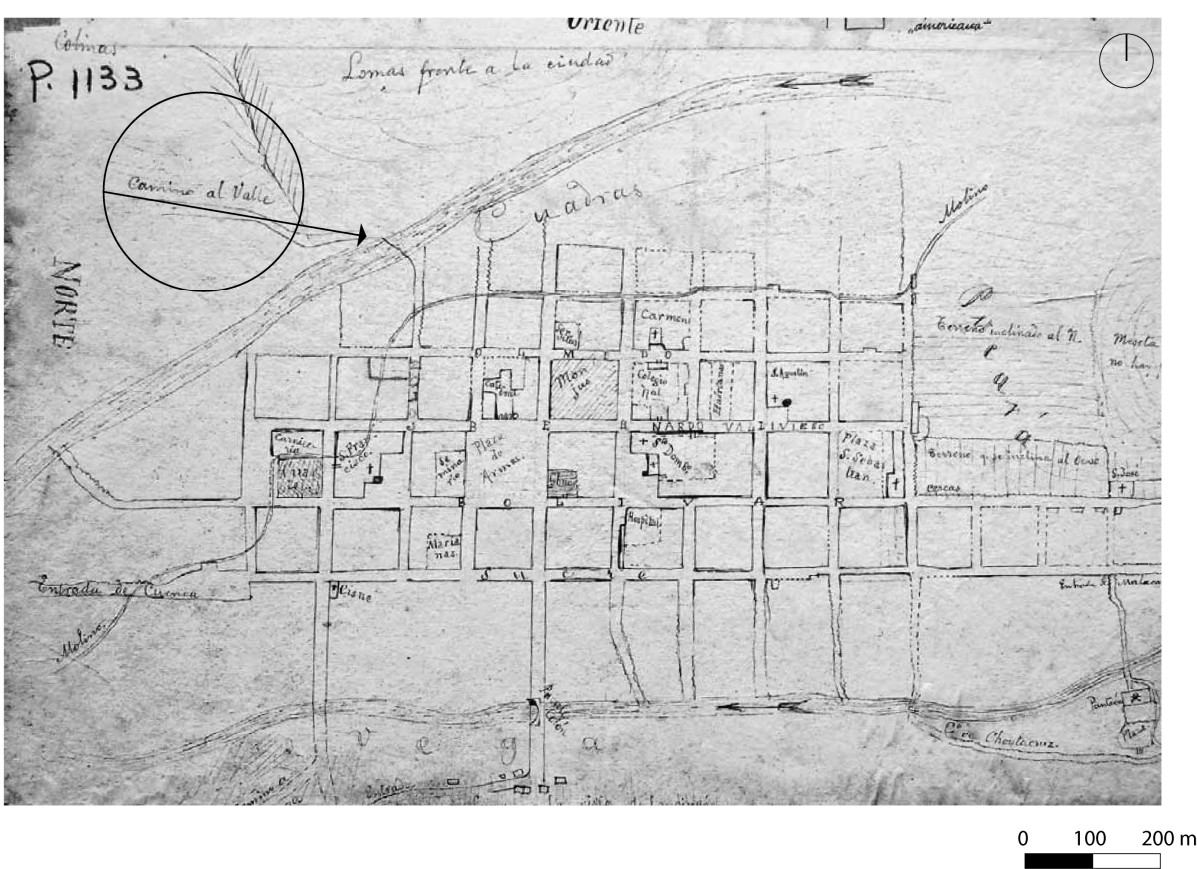

**Figure 2.** Map of Loja in 1880 [40]. At this date, the city was not yet consolidated, and the road leading to the Valley could be individuated.

Furthermore, the implementation of the Agrarian Reform Law, approved in 1973, imposed requirements such as the efficient utilization of over 80% of the land to avoid expropriation. It mandated a level of productivity equal to or greater than that established by the Ministry of Agriculture and Livestock and outlined grounds for expropriation, including the existence of non-wage relations and demographic pressure. The immediate effects of this law expedited land distribution processes and, although it did not establish maximum limits for landholdings, it introduced the concept of 'land grabbing' [41], par-

ticularly prevalent in the inter-Andean region where latifundia (large agricultural estates) were common. Indirectly, this policy contributed to a lack of employment opportunities in rural areas and fueled rural–urban migration.

The absence of adequate public policies, difficulties accessing official housing programs, and the prevalence of poverty led rural populations to occupy the slopes of San Cayetano. Until 1990, the sector remained predominantly rural, lacking basic services and relying on water supplied by nearby streams. It was not until 1998 that the state system finalized the road connection to other provinces, including the road to Zamora and the route Oriental de Paso.

This development marked the beginning of settlement in the area, initiating a process of appropriation [42]. Additionally, in the same year, Ecuador experienced an economic crisis that prompted the implementation of dollarization, further exacerbating waves of internal and external migration.

## 3. Results

### 3.1. Consolidation Process in San Cayetano: A Superposition of Spatial Fragments

Between 2009 and 2019, San Cayetano consolidated 65.29% of its territory within the urban boundary. Due to the legalization process, the State began to provide basic infrastructure services and public facilities such as stairways, health centers, public spaces, and asphalted roads [42]. According to the last available Census (INEC; 2010), in 1998, the population reached 5513 inhabitants and is projected to reach 8435 inhabitants by 2030.

According to the Land Use and Management Plan 12, the parish of El Valle comprises 7424 land-use units, which account for 13% of the total urban area (Figure 3). The "non-urban and special" land use is the most prevalent, encompassing 3901 units (representing 53% of the total) [36]. This dominance is attributed to the prevalence of agricultural activities in small plots, poultry farms, and numerous vacant lots that remain undeveloped due to geological and morphological constraints.

The urban fabric consists of 57 blocks containing 1454 lots, which exhibit variations in size and shape. Smaller lots are typically situated in lower areas and in proximity to roads or urban facilities, while larger lots are found farther away (Figure 4). The formal road connection system is deficient, with only 60% of roads featuring asphalt covering, sidewalks, and curbs, while the remainder is composed of ballasted roads. Additionally, informal roads such as paths and stairways exist to shorten distances.

Analyzing the current state of consolidation within the neighborhood, it becomes apparent that a more consolidated nucleus exists in the southeastern sector, where the main public space and the church are situated. This sector is characterized by taller buildings, which have replaced earthen dwellings and tend to occupy the maximum available area within the lot. From this central area, consolidation gradually diminishes radially, leading to territories that are progressively less consolidated, often serving agricultural purposes.

The neighborhood is significantly influenced by the pronounced topography; this factor has a relevant impact on the level of consolidation. In the lower part of the settlement, near the main road connecting to the city center and where the slope is the least steep, there is a high concentration of services, infrastructure, and public spaces. Consequently, this area presents an urban image closely integrated into the urbanization process. It is worth noting that initial observations in San Cayetano, as part of academic activities developed by the authors, date back to 2018, and within six years, significant improvements have been observed in this sector.

However, despite the acceptable level of consolidation in this portion, the spontaneous nature of the fractionation process is evident in the complex network of connections providing access to different properties. The integration of the neighborhood into the formal city has not addressed deficiencies in the public infrastructure system, which has been developed incrementally and is currently unable to adequately meet the demands of the growing population.

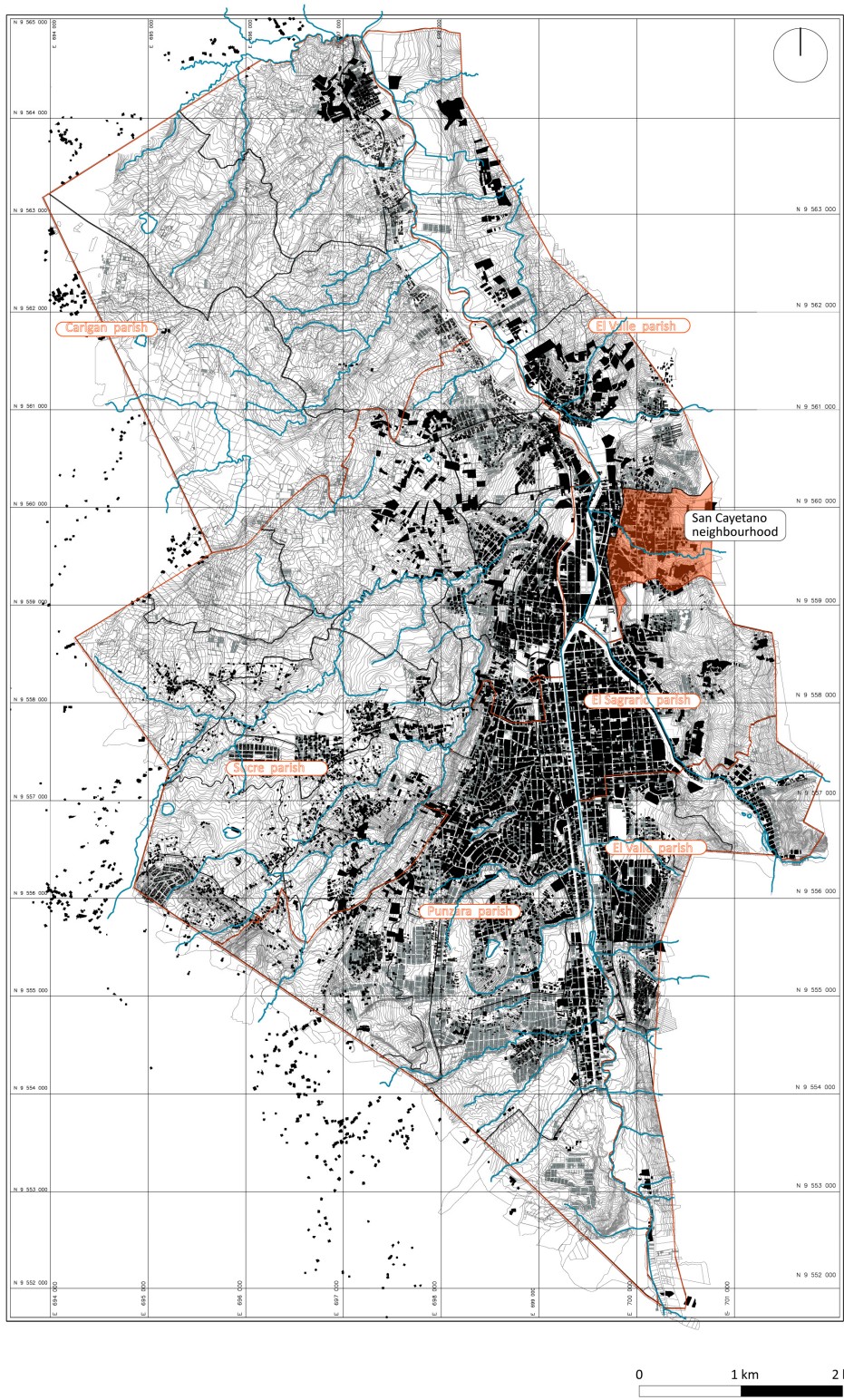

**Figure 3.** The current shape of Loja City (2020). Source: Municipal cartography—department of urban planning, Municipality of Loja. The parish of El Valle is a part of the city of Loja, which consists of six urban parishes in total. The neighborhood of San Cayetano is located on the northwestern border of the city, within this parish, occupying a differentiated area within the broader context of Loja. Edition: the authors (2024).

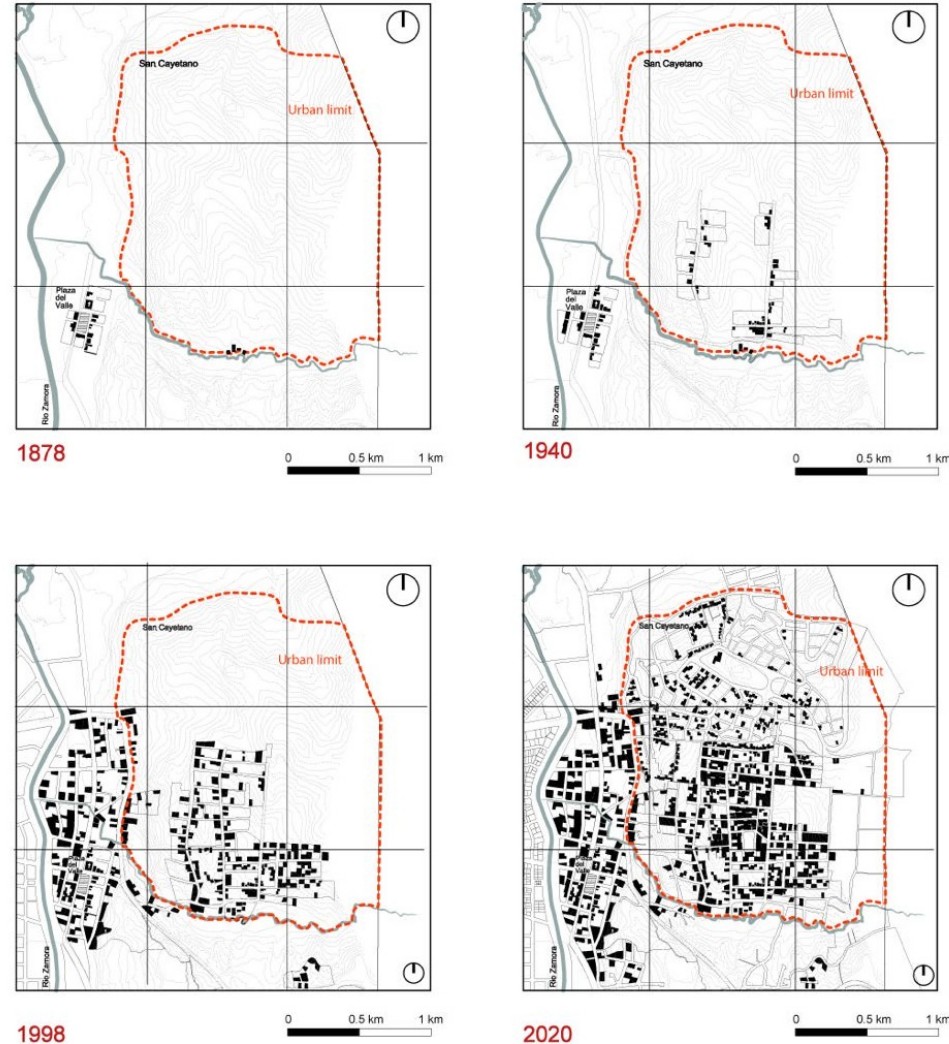

**Figure 4.** The development process of the San Cayetano neighborhood from its beginning in 1878 to its progressive consolidation, as evidenced by the progressive occupation of the hillside for urban expansion. Source: the authors (2024).

In this regard, it is evident how the development of infrastructure in the area occurs progressively, often as a result of gradual appropriation processes. This has led to the establishment of a complex network of streets, stairs, roads, paths, intersections, and barriers, highlighting the diverse range of uses that these urban elements serve.

As one moves towards the peripheral areas of the neighborhood, the degree of consolidation progressively diminishes, leading to spontaneous architectural solutions and an increasingly pronounced lack of public infrastructure. However, analyzing the settlement formation process allows us to understand how these two realities represent different stages of the inexorable evolutionary process characterizing peripheral environments.

In the upper parts of the neighborhood, the first signs of urbanization are becoming visible through the construction of a few residential complexes and the opening of new roads, which pave the way for further expansion possibilities. Although the sector's integration into the urban economies of the neoliberal city has seemingly facilitated rapid expansion, it is evident that this expansion continues to occur precariously. Interventions that are indifferent to the context contribute to an image of urban sprawl, lacking sensitivity to the unique characteristics of the area.

Simultaneously, the configuration of the parceling reveals potential future expansion areas within the neighborhood. In the northern sector, an urban structure characterized

by regular subdivisions is present, although it has not yet been fully integrated into the city. In any case, analyzing both occupied and vacant spaces within the perimeter allows for the recognition of a neighborhood in a phase of consolidation. Significant portions of the territory are still governed by the natural system, indicating ongoing development and potential for further urbanization (Figure 5).

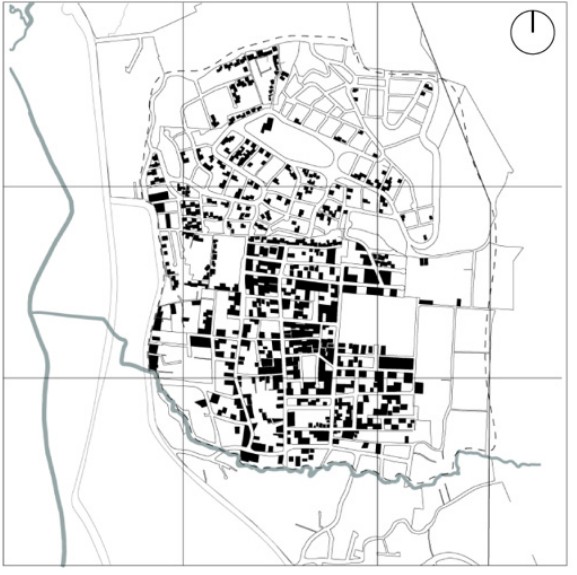
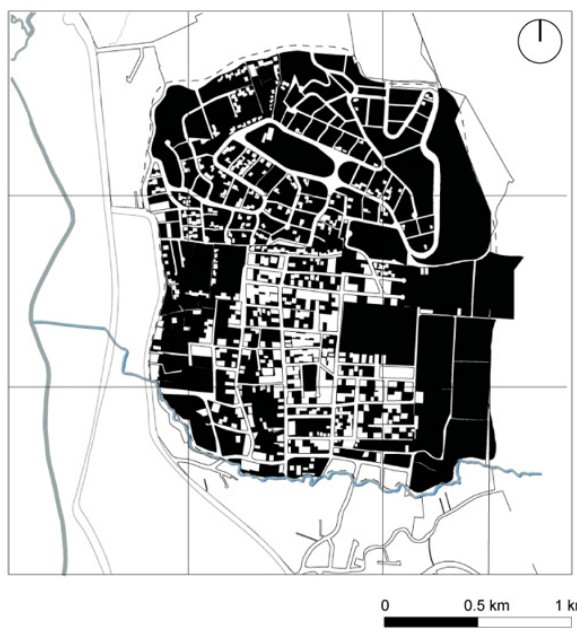

■ Built lots                    ■ Un-built lots

**Figure 5.** Built and unbuilt plots in the consolidation process of the San Cayetano neighborhood. Occupied and vacant spaces can be identified in the area, with the central area of the neighborhood showing a higher concentration of built-up plots, compared to the year 2020. Source: the authors (2024).

The process of urbanization in the district is driven and facilitated by the rapid rise in land prices, which have experienced a significant surge in profitability over the past fifteen years. According to data from the Department of Appraisals and Land Registry of the City Hall, the value per square meter has increased by an average of 463% in the parish of El Valle from 2008 to 2022, with an average annual variation of 39%.

The analysis of socio-economic indicators reveals a precarious reality, which is reflected in the condition of the physical infrastructure. The poverty rate in the parish of El Valle reaches 9.4%, with over 30% of the population engaged in informal employment or lacking empowerment [43]. The average monthly income for residents of the parish is approximately $549 for men and $179 for women, indicating a significant disparity compared to the central parish of the city, where incomes are nearly double [24]. Moreover, there is a notable gender wage gap, suggesting a family model reliant on a single income. These socio-economic conditions significantly influence the urban and architectural landscape of the neighborhood, resulting in a diverse array of spatial production practices.

The fieldwork found evidence that the neighborhood is characterized by rural activities associated with primary agricultural endeavors. Many plots serve as spaces for farm animals and cultivation for self-consumption, intermingled with production activities such as food stalls, small carpentry shops, and grocery stores, among others.

Regarding its architecture, the neighborhood exhibits a mixture of single-family houses and larger residential buildings. There is a gradual vertical progression evident in the larger buildings. The materiality of the structures varies, ranging from traditional houses constructed with earth, in minor parts, which have been integrated into newer constructions, to dwellings with irregular setbacks and fragile materials such as boards, cardboard, and

zinc, among others. These structures are notable not only for their design but also for their dimensions and materials.

To provide a reading of the spatial composition of the neighborhood, it is important to superimpose the analysis of factors such as topography, the structure of the road system, and parcellation, understanding how the urban process faces the challenges imposed by the morphology of the territory.

The San Cayetano neighborhood stretches across a slope, encompassing areas with medium to steep inclines. The layout of roads is heavily influenced by the topography, resulting in two main configurations of traffic axes. In the central more consolidated sector, a regular grid pattern predominates, although some roads within blocks may have been spontaneously established and later stabilized. In the less urbanized sector, where slopes are steeper, the geometry of the road layout becomes more sinuous and is more sensitive to the natural conditioning factors of the land (Figure 6).

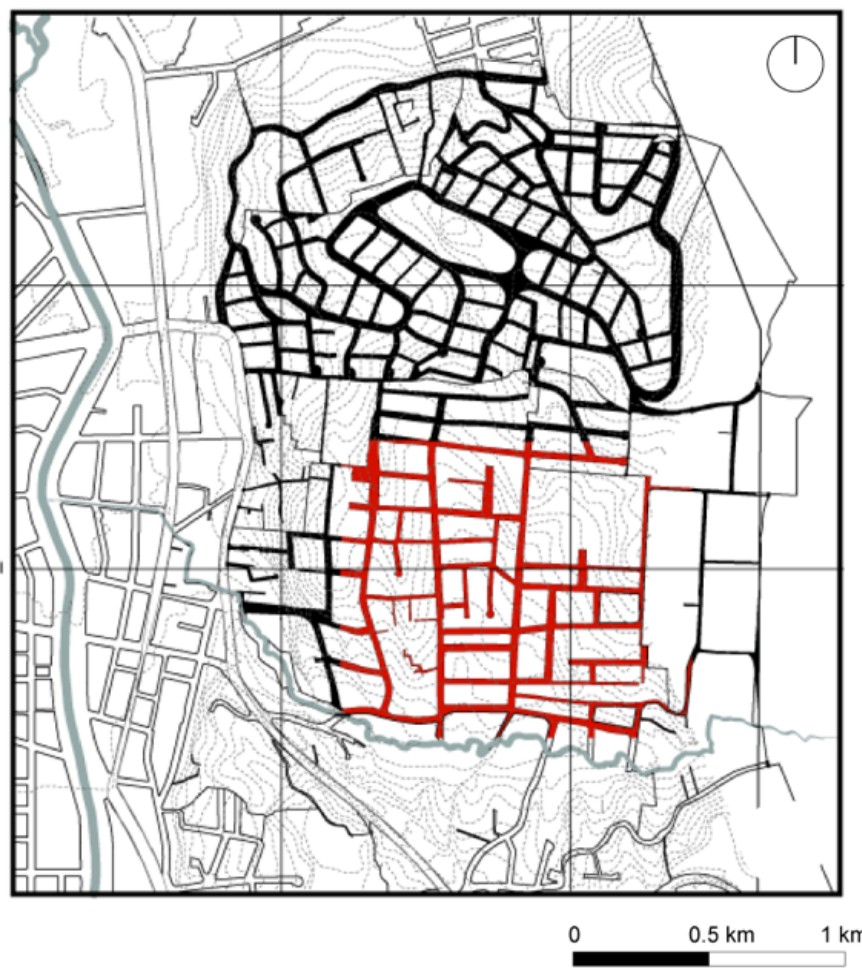

**Figure 6.** The road path in San Cayetano. Relationship between road path, topography, and consolidation. Two situations are distinguished: one corresponds to a regular layout that characterizes the central core of the neighborhood and the other more irregular plot, which characterizes expansions on the slopes. The cartography provided is from the year 2020. Edition: the authors (2024).

Based on the neighborhood-scale analysis, three block typologies can be identified, distinguished by their consolidation status and average size. This reflects the heterogeneity of situations within the San Cayetano neighborhood. The largest blocks are primarily utilized for cultivation and livestock, often lacking traces of urbanization. Meanwhile, blocks ranging between 2000 and 5000 m² are frequently subdivided into smaller plots or are undergoing a similar process of division (Figure 7).

**Urban layout**

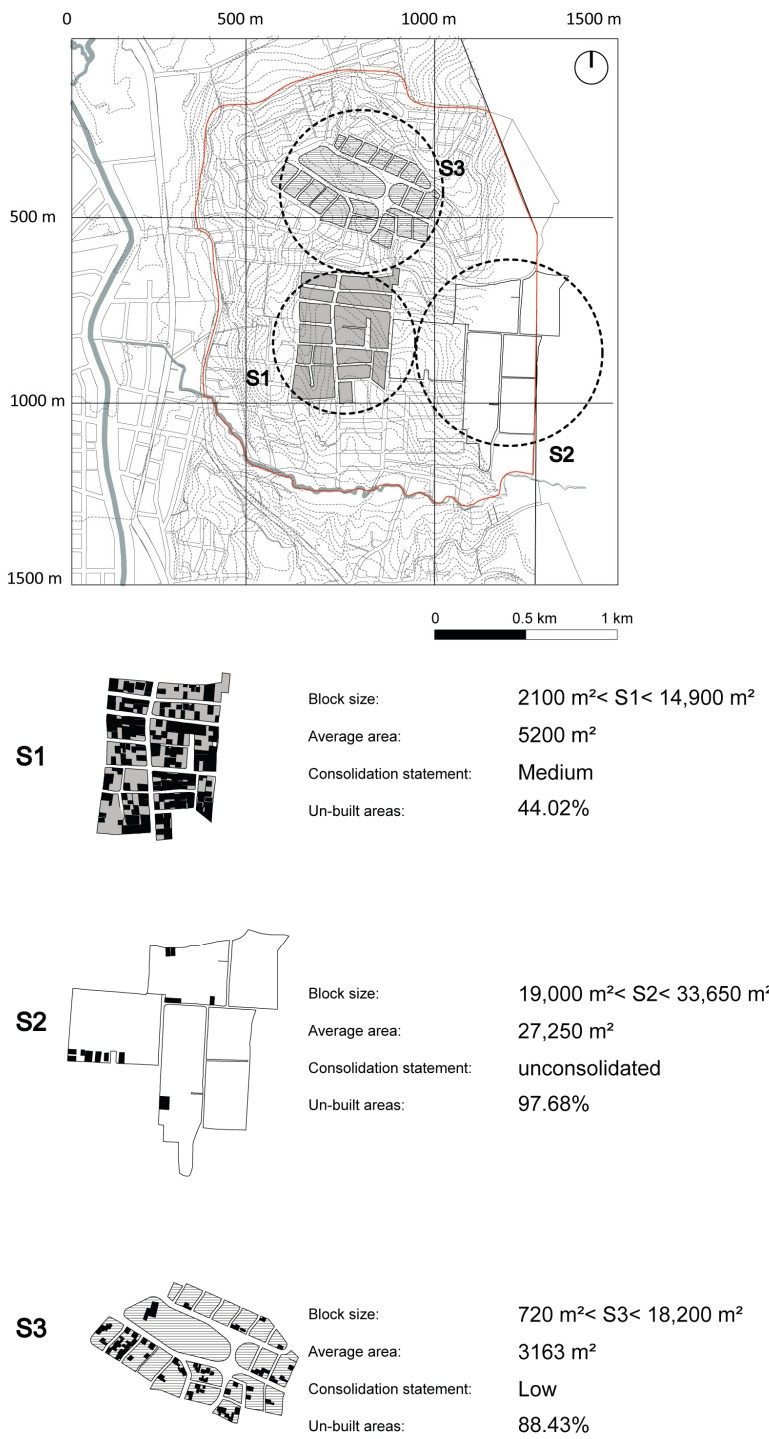

**Figure 7.** Urban layout. Differences in size and consolidation status of lots in three sectors of the San Cayetano neighborhood. The cartography provided is from the year 2020. Elaborated and edited by the authors (2024).

### 3.2. Appropriation Phenomena in the Fragmented Space

The study of the San Cayetano neighborhood yields unique insights that contribute to the understanding of the settlement's morphology and can be generalized to identify spatial patterns characteristic of peripheral environments in Andean intermediate cities. Upon initial analysis, it becomes evident that traditional interpretations, which rely on a center-periphery

dependency approach, are insufficient for comprehending the multifaceted nature of these areas. Currently, the lack of conceptual frameworks for defining rural–urban interfaces underscores the importance of this study, prompting ongoing research in environments where architectural and urban forms are inherently evolutionary.

The observation of the San Cayetano neighborhood allows us to classify this territory as a spatial diaphragm that integrates urban and rural living practices. Within this context, diverse realities coexist, resulting in a kaleidoscopic and fragmented image characterized by ingenious individual solutions that collectively form the urban mosaic (Figure 8).

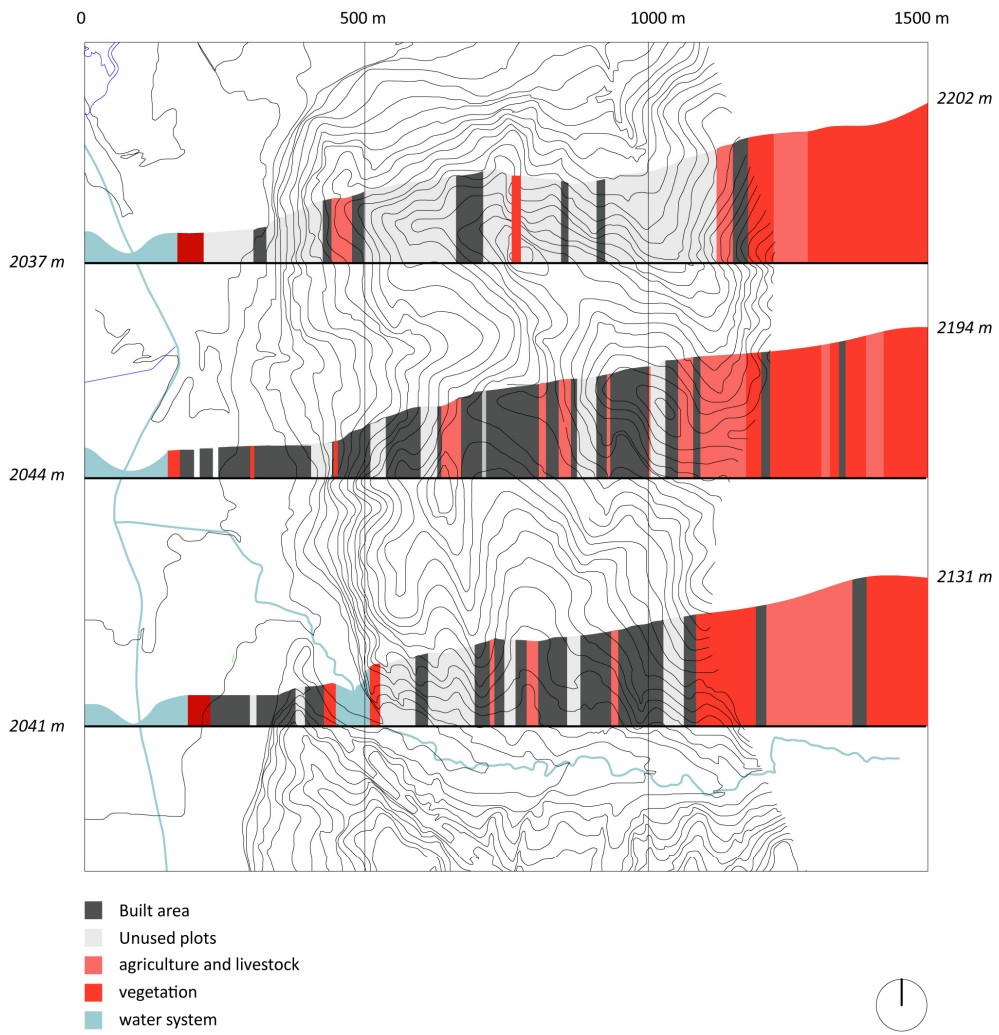

**Figure 8.** Topographical and land use sections. The alternation of uses between rural and urban can be observed in the topographical section of the San Cayetano district. Source: the authors (2024).

The peri-urban edge is not defined by a clear boundary; rather, it represents a spatial situation characterized by heterogeneity and fragmentation, resulting in a landscape of continuous change. Within this context, agricultural fields, built environments, and residual areas coexist within the same space, often lacking any discernible relationship between them [14]. The reality encapsulates the dualities and contradictions that arise from the intersection of urban and rural worlds, reflecting the cultural characteristics of a population transitioning into urban economies while retaining ties to its agricultural origins. Within this context, it is common to encounter predominantly rural practices within the area, such as cultivation and animal husbandry, alongside the presence of urban services like mechanics, supply points, and product sales.

The socio-economic factor plays a crucial role in shaping these inhabiting practices. Often, the necessity to augment family income prompts the activation of informal economies within the gaps and interchanges of the neighborhood. These services often exhibit a temporary nature and are conducted within structures characterized by precariousness, set up either within the household spaces or within the confines of the lot.

Services often develop by occupying the front portion of the building, thus orienting the residence towards the street and changing the use of this area of the land. The front setback, although legally a private space, frequently assumes semi-public connotations, allowing for the reception of customers for various services.

The ongoing growth and increasing urbanization of San Cayetano are gradually displacing the traditional agricultural practices to the less consolidated parts of the area. In these areas, the presence of orchards may still be observed, although they are becoming less prevalent as urbanization continues to expand. In the more peripheral areas, moreover, it is common to find activities such as breeding, often managed at a family level (Figure 9).

The analysis of architectural forms has confirmed the kaleidoscopic image of the neighborhood. In the more consolidated part of the neighborhood, there is a proliferation of high-rise buildings accommodating both housing and businesses. Conversely, in San Cayetano Alto, smaller structures are prevalent, often constructed precariously. This is due to the constant need to adapt to the sloping terrain. Considering that the landscape has been gradually intervened by property owners, each building exhibits a unique relationship with the street, and features sloping access.

In this regard, the neighborhood landscape is both constant and in a state of perpetual transformation, resembling a collage of architectures that appear to belong to different eras and places. Approximately 4% of the structures are traditional houses constructed with earth, while 7% are made of concrete. Additionally, 10% exhibit a diverse array of materials such as scrap wood, cardboard, tin, and plastic, with brick being the predominant material in 10% of the buildings.

The urban image is impacted by various factors, including both formal and informal connections. For instance, the road layouts planned by the municipality may not account for rural housing, particularly those structures of older vintage that are situated at different elevations compared to the projected road layout. Often, the new road connections opened to enhance accessibility to the gradually built lots have had to contend with both topographical conditions and land ownership issues. In San Cayetano, as in many peripheral settlements, residents often resort to the mechanism of "Paso de servidumbre," which involves connecting their plot of land with the public road by opening a path that traverses neighboring properties. This is accomplished through a process of land cession by the different property owners. While these initiatives may have a legal framework regulated by the municipality, they are often carried out through informal agreements among residents.

Additionally, natural paths and improvised stairways serve to shorten distances, altering natural spaces and widening inequalities within the sector. The appropriation of spaces to convert them into public areas for social, economic, and recreational activities results in landscapes that are contradictory.

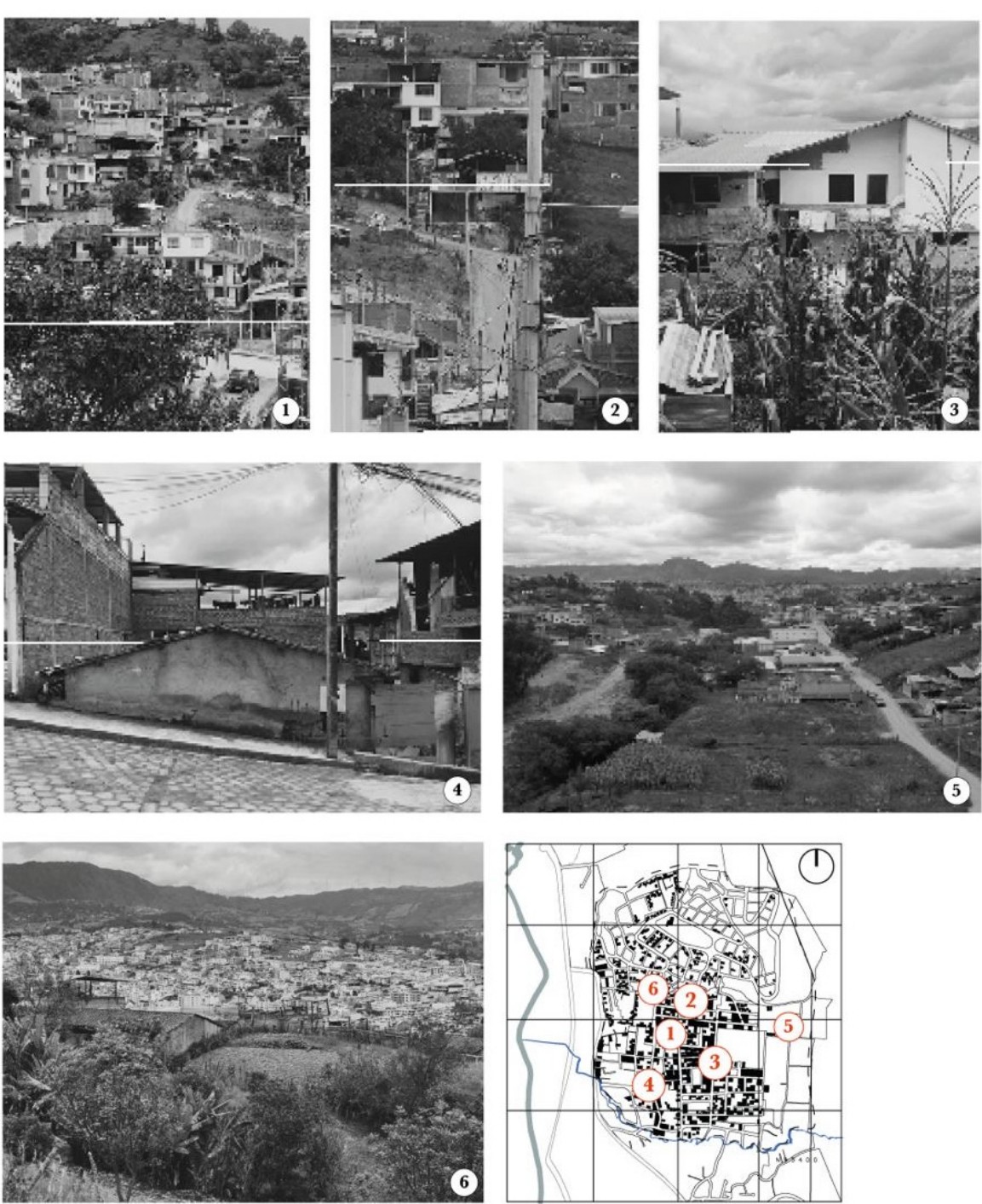

**Figure 9.** A selection of photographs captures various aspects of the neighborhood, showing different facets of the peripheral landscape, characterized by the intersection of practices from rural and urban environments. This superimposition of apparently contrasting elements creates a dynamic and heterogeneous landscape, such as dwellings with different materiality and a combination of primary practices such as farming. Source: the authors (2024).

## 4. Discussion

By analyzing the territory of San Cayetano, it is possible to identify the predominant characteristics of spatial fragmentation, creating a mosaic-like appearance typical of the contemporary periphery, where fragments are intermixed yet distinct. This study is based on the hypothesis of a spatial occupation that defies strict categorization as either rural or urban, lacking a precise definition.

Niembro [44] highlights that numerous quantitative and segregation indices tend to concentrate solely on specific dimensions or variables, often resulting in binary outcomes like center-periphery distinctions. Nevertheless, by incorporating the fragment dimension, we can transcend the binary paradigm frequently employed in such definitions, acknowledging the present heterogeneity characterizing peripheral circumstances.

The rapid transformation from rural to urban areas has significant implications for the general morphology of the neighborhood. This transformation results from a process of agglutination and overlapping of elements, both in physical and spatial terms, as well as social and cultural aspects.

This trend is common in various other South and Central American contexts, which can be compared to the case study as they are located on the periphery of rapidly growing intermediate cities. Indeed, as highlighted by Klaufus [45], it is common to encounter suburban disorder attributed to fully serviced residential projects for a new, middle class, built-on, ecologically vulnerable land.

While a collective urban image is still emerging, inhabitants are gradually developing a sense of belonging. However, the migration of low- and middle-income groups, driven by limited resources, sees peri-urban areas like this neighborhood as ideal locations for personal housing development, resulting in increased land consumption.

This trend is substantiated by the analysis of per capita income and employment status, revealing that limited resources allow for only incremental development. Indeed, the precarious employment situation coupled with low wage levels only allows for occasional measures to improve the house. In this process, the consolidation of physical infrastructure is largely undertaken independently by inhabitants, often resulting in increased fragility of the building stock. Consequently, one can observe a population gradually attempting to integrate into the formal dynamics of the city, yet remaining partially connected to rural production practices. This connection is exemplified, for instance, by small-scale cultivation in open spaces within residential lots. Despite being considered an urban neighborhood, San Cayetano still exhibits mixed uses intertwining urban and rural characteristics.

Furthermore, the land of San Cayetano represented a significant source of income for the owners, who saw the opportunity to subdivide the plots to initiate real estate activities. This activity has also paved the way for real estate companies, which have been able to purchase land at very advantageous prices, turning the soil into a commodity. This is evidenced by the reduction of the minimum lot size, which is particularly noticeable in the more consolidated areas.

The speculative process of land commodification reveals, as in other case studies, a significant proportion of agricultural land use in areas at risk of urban expansion [46]. In this regard, Haller [47] emphasizes the importance for planners and policymakers to address the risks associated with the risk to the depletion of agricultural land in developing peri-urban regions.

The speculative process is confirmed by the fact that, despite the neighborhood having high levels of poverty, there has been no initiation of social housing production.

However, despite the rapid process of speculation highlighting the social disparities it generates, in the case of the San Cayetano neighborhood and generally in the city of Loja, these disparities are not directly linked to the peripheral gated community urbanization. Instead, this type of urbanization is evident as a recurring pattern in similar contexts. Therefore, there is not an evident division between the open city and the private one as two parallel cities, denoted by Klaufus [45] in the Central American case.

The urban form of the neighborhood reflects the overlapping of societal criteria, such as the imposition of an orthogonal grid on challenging topography, resulting in forced landscapes and constructions that must contend with these difficulties. As Niembro [44] points out, in the case of Bariloche, another notable factor contributing to the discontinuity and dispersion of the urban area is the physical environment itself and the existence of natural barriers that influence urbanization patterns.

Unplanned processes of excavation and filling of slopes further alter the landscape and increase the vulnerability of buildings, regardless of the materials used. This results in irregular and multifarious plots, as well as fragile buildings, contributing to a discontinuous and heterogeneous urban image.

The vulnerability of the building stock is evident across various analytical scales, taking into account factors such as building location, architectural morphology, habitability standards, and the proper use of construction materials. Buildings are often situated on steep slopes without appropriate structural reinforcement systems, and they lack proper management of sanitation infrastructure and waste disposal. It appears that buildings attempt to carve out space, seeking favorable conditions for establishment, but without considering the limitations imposed by the slope.

With a desire to maximize available space, constructions often take on an incremental character, resulting in internal divisions aimed at increasing room numbers, sometimes at the expense of proper lighting and ventilation conditions. Additionally, the evolution of construction provides insight into the consolidation process. For instance, in multi-story buildings, the consolidated lower portions typically feature reinforced concrete and brick or block infills, while upper floors may consist of exposed terraces or unfinished areas lacking fixtures, finishes, and flooring.

The use of various materials, assembled at different times and corresponding to different extensions, is often undertaken by individuals lacking technical expertise. Consequently, construction errors are common, compromising the integrity of the building.

Furthermore, the gradual construction of the neighborhood has resulted in the emergence of public spaces that are essentially byproducts of the urbanization process. These spaces are often difficult to delineate from private areas, contributing to a sense of uncertainty among residents. The presence of narrow alleys and poorly lit passages complicates social control over public spaces, leading to decreased activity within the neighborhood during evening hours and fostering safety concerns. After sunset, public gathering spots remain largely vacant, except for the central space, which accommodates a volleyball court and remains the most frequented meeting point within the neighborhood. Identified mono-functional and low-density usage patterns contribute to spatial and social segregation, as well as a strong dependence on services from the city center. The spatial analysis of San Cayetano in this regard shares similarities with that provided by Rojas [48] on the metropolitan area of Concepción in Chile, where it is highlighted that urban sprawl indicators indicate that areas with higher levels of development will exhibit greater complexity but reduced compactness, particularly in the larger and most urbanized areas.

While the analysis of space appropriation practices suggests the progressive emergence of small-scale neighborhood services such as grocery stores, mechanics, and craft shops, the neighborhood remains unequipped with essential services required for autonomous development.

It is important to emphasize the substantial disparity between official cartographic documentation, regulatory guidelines provided by the municipality, and the actual configuration of the study territory. Indeed, the comparison between existing cartography and satellite imagery or field survey data often reveals numerous discrepancies. These include buildings constructed in non-designated areas, accessibility solutions implemented outside permitted regulations, and internal plot occupations that disregard current building codes.

This inconsistency is partly attributed to the lack of updates in maps and quantitative data, which fail to accurately depict the reality on the ground. Additionally, the informality inherent in the neighborhood's construction process, often legalized retroactively following architectural interventions, contributes to this discrepancy. Updating the tools for analyzing the territory is considered paramount as they provide a comprehensive understanding of the neighborhood's reality, serving as the documentary foundation for any actions aimed at enhancing existing infrastructure.

## 5. Conclusions

The liberalization of the land market has led to the commodification of each plot of land, fostering speculative processes. In Ecuador, as in other regions of South America, horizontal and peripheral growth trends have continued since the 1990s, coinciding with the strengthening of neoliberal models in the continent.

Rural areas, historically tied to haciendas, have undergone progressive subdivisions based on the principle of maximizing profitability. Large plots allocated to beneficiaries during the agrarian reform have been subdivided into smaller parcels, typically without adherence to municipal regulations but rather adjusted upon regularization.

Indeed, the increasing profitability of land for urban development compared to its agricultural use often leads to the transformation of undeveloped areas into urban spaces. This trend reflects the dynamics of real estate markets, where developers seek profitable opportunities in peri-urban areas. As a result, the landscape and spatial dynamics of these areas continue to evolve, influenced by economic forces and urbanization trends.

The speculative process indeed underscores a widening gap between the natural and the anthropic systems. The imposition of rigid urban grids on sloping terrain reflects a tendency to disregard the inherent characteristics of the landscape. The rapid pace of urbanization has intensified pressure on soil resources, driven by new urban developments and the regularization of informal settlements gradually integrated into municipal service networks. This heightened anthropic pressure on the land is a defining feature of peripheral expansion in intermediate cities, reflecting common patterns observed nationally and internationally.

Among those, the system's inability to adequately respond to high population demands, the absence of effective public policies, hyper-specialized urban sprawl, environmental degradation, and limited access to services for low-income individuals are the primary challenges faced by these cities. Addressing these issues is crucial for ensuring sustainable urban development and improving living conditions for all residents.

Regulating land use in peripheral areas is indeed crucial for curbing the uncontrolled proliferation of urban sprawl. Municipal authorities play a key role in implementing measures to address the causes of accelerated and uncontrolled urbanization. By establishing clear regulations and guidelines for land use, authorities can steer development in a more sustainable direction, ensuring that growth is managed in a way that preserves natural resources, promotes equitable access to services, and fosters resilient communities. Effective urban planning strategies, such as zoning ordinances, land use restrictions, and incentives for compact development can help guide growth in peripheral areas while protecting sensitive ecosystems and promoting balanced urban development. Additionally, engaging stakeholders and including local communities, developers, and environmental organizations in the planning process can lead to more inclusive and participatory decision-making, ultimately contributing to more sustainable urbanization outcomes.

Effective municipal action to regulate urban expansion and diversify land use is paramount for combating urban sprawl and promoting sustainability in intermediate Andean cities and similar urban areas worldwide. This approach demands a collaborative effort involving local authorities, communities, and various stakeholders engaged in urban development. By working together in a coordinated and multidisciplinary manner, these entities can implement policies and initiatives that foster fair, resilient, and environmentally friendly urban growth. Through careful planning, thoughtful land use management, and inclusive decision-making processes, cities can achieve more sustainable and equitable development outcomes that enhance the quality of life for all residents while safeguarding the natural environment for future generations.

Finally, as this is the first part of an ongoing investigation, it is necessary to carry out complementary studies to obtain data from other areas, such as geological and hydrological studies, to identify potential risk areas so that land use can be regulated. Likewise, environmental impact studies should be considered to evaluate the alterations of the constructions to the natural environment. Finally, a socioeconomic analysis should be carried out to

evaluate the impact of the infrastructure on the local community. These additional studies will guarantee a sustainable and safe development of hillside constructions.

**Author Contributions:** Conceptualization, V.D. and C.C.; Methodology, V.D.; Validation, K.M. and C.C.; Formal analysis, K.M.; Investigation, K.M., V.D. and C.C.; Resources, K.M.; Writing—review & editing, V.D.; Visualization, C.C. All authors have read and agreed to the published version of the manuscript.

**Funding:** The paper received no external funding.

**Data Availability Statement:** Data is contained within the article.

**Conflicts of Interest:** The authors declare no conflict of interest.

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
