# Peer review of "The Formation of the Urban–Rural Fringe Space in the San Cayetano Area: The Transformation of a Peripheral Urban Landscape in Ecuador"

_land, doi:10.3390/land13040494_

Round 1

Reviewer 1 Report

Comments and Suggestions for Authors

Figure 1 looks grainy - check

San Cayetano → to be located well on the map

It would be good to do a geographic framing with a figure where there is also a base map (e.g. orthophoto, openstreetmap etc),e.g. improving figure 3 (however you have to enter the print scale of the figure: scale bar and/or numerical)

The analysis of urban and rural growth history is well described.

Some figures are without captions.

In Figure 5 -->Portuguese Lotes edificados

Line 224 Not --> Note

Line 232 Character formatting

Does Figure 5 refer to the year 2020? specify.

It would be useful to quantify the transformation phenomena and spatial data, which are analyzed, including through GIS spatial analysis (e.g., land consumption, fragmentation, etc.).

Author Response

The title of the manuscript changed from " Analysis of the urban peripheries of the Andean intermediate city and its consolidation process. The third space of Barrio San Cayetano, Loja.
Now: "The formation of the urban-rural fringe space in the San Cayetano area: the transformation of a peripheral urban landscape in Ecuador"
All incorporations and suggestions are marked inside the text in gray.
Thank you for your time and recommendations.

Reviewer 2 Report

Comments and Suggestions for Authors

The manuscript examines a useful issue, and the topic of research is relevant in my opinion. However, there are quite a few errors in content and form in the manuscript. I think these are as follows.

The title of the manuscript does not fully overlap with the contents of the manuscript. The title should be more specific and Ecuador should be mentioned, for example, "The formation of the urban-rural fringe space in the San Cayetano area: the transformation of a peripheral urban landscape in Ecuador".

The abstract does not mention the main results and conclusions of the research. This should be added at least in 2-3 sentences.

The text of the manuscript in its present form is short, I think there is still plenty of scope in the journal, so there is room for the theoretical addition I propose below.

The literature in this study is exclusively in Spanish and Italian, which is not appropriate for a global journal, as the majority of global readers may not be able to access or translate these works due to language and other limitations. For this reason, English sources should also be provided (and, in addition, titles of non-English works should be given translated in parentheses after the original titles).

Please tell the global reader more about which academic field you are (mostly) interested in, and from which point of view you study the highly interdisciplinary topic you have chosen. Architecture, cultural anthropology, urban history, urban studies, or urban development? Because I do not suppose it is clear from the manuscript at the moment.

The authors apply and often mention their concept of “third space”. I would like to draw your attention to the fact that the concept of Soya's “thirdspace” is widely known in urban studies, who used it before the authors, but in a different sense (see, e.g., Soja E. W. 1998). However, the authors do not mention anything about him in connection with “third space” in the manuscript. However, I think this can be confusing for the reader, so the authors should address this overlap in their theories section and clarify the difference between the two "third space" concepts.

I do not suppose that the authors explicitly mention why they chose the city and its neighbourhood within Ecuador. Please give more objective arguments for this, or, if there are none, write more about subjective factors to see if they are scientifically acceptable.

In the theoretical part, I think that the authors write very little about whether the phenomena analysed in the city and neighbourhood they studied (e.g. the "third space" they defined) exist in other cities in Ecuador or in other countries in Latin America. However, as far as I know, they exist in other cities in Ecuador and other countries of Latin America, moreover, the mixing of urban and rural features described by the authors occurs in many other urban suburbs of the "global South", partly in a similar way. For this reason, the author should refer to such English-language literature as well as I have mentioned. In connection with this, they should briefly discuss how much the phenomena examined by them are similar and different compared to other cities in Ecuador and other countries of Latin America, and partly compared to global characteristics. Obviously, the authors do not intend to make comparisons, which is understandable, but they should include all the comparisons I have requested in the manuscript. Otherwise, the authors' manuscript, compared to the generality of the subject they chose, may seem somewhat self-serving and narrow-scoped. And, such a comparative additional analysis would decrease the highly descriptive and, in my view, less analytical character of the manuscript.

There are also relatively few references in the manuscript to research or theories other than those I suggest that could help readers understand the context of the research.

Methodological information seems to be incomplete. Please specify exactly how long you have used these methods in time.

In a global journal, it is always worth using a map showing the location of the city chosen by the authors within Ecuador. And global reader also needs a new map that shows the better location of the neighbourhood under study within the city. Most of the maps are barely visible due to poor resolution. For this reason, please increase the size of the maps (there is plenty of space for this in the journal). Please have titles (captions) for all maps. All maps should have scale bars and only English text, not Spanish.

The Photos are basically worthy, but you will not tell them as much if you do not have titles (captions) to indicate what your purpose is with them. And at the moment it is not clear where these were taken in your study area, so please mark them on one of the maps (e.g. in the form of numbering).

I assume that the Discussion and Conclusions sections are disproportionately short compared to the rest of the journal and should be expanded. For example, it would be possible to mention in much more detail what the authors propose on the matter, and how the disadvantages described in their manuscript can be reduced by local politicians, planners or NGOs. I suppose this part of the proposal would be a predisposition to the choice of topic. The Discussion and Conclusions sections (and other parts of the manuscript) do not really contain research limitation-type notes, although they are usually present in all research. The authors should write more about these. And to what extent do the authors think the results of the study can be generalized to other cities or neighbourhoods within Ecuador or Latin America?

grammar:

I mention some errors that need to be fixed:

"the urban form" (row 113)

different font types (rows 232-234)

Increased sentence complexity. Some sentences are too complex and cumbersome, making them difficult to read and understand, and should be simplified. Example: "The immediate effects of this law expedited land distribution processes and, although it did not establish maximum limits for landholdings, it introduced the concept of 'land grabbing', particularly prevalent in the inter-Andean region where latifundia (large agricultural estates) were common." (rows 184-188)

Comments on the Quality of English Language

I mention some grammar errors that need to be fixed:

"the urban form" (row 113)

different font types (rows 232-234)

Increased sentence complexity. Some sentences are too complex and cumbersome, making them difficult to read and understand, and should be simplified. Example: "The immediate effects of this law expedited land distribution processes and, although it did not establish maximum limits for landholdings, it introduced the concept of 'land grabbing', particularly prevalent in the inter-Andean region where latifundia (large agricultural estates) were common." (rows 184-188)

Author Response

(The authors gave the same response as above.)

Reviewer 3 Report

Comments and Suggestions for Authors

The manuscript titled "Analysis of the urban peripheries of the Andean intermediate city and its consolidation process. The third space of Barrio San Cayetano, Loja" provides a comprehensive exploration of the spatial dynamics and consolidation processes within the peripheral neighborhood of San Cayetano in the city of Loja. The abstract succinctly outlines the study's focus on the relationship between socio-economic dynamics and urban configuration in the Andean context. The author employs a mixed-methods approach, combining official data with on-site visits, cartography, and graphic representation to identify recurring spatial configurations that characterize the third space of the Andean periphery.

The introduction contextualizes the study within the contemporary urban phenomenon of peri-urbanization, emphasizing the emergence of a third space that blurs the traditional boundaries between urban and rural areas. The historical and contextual background provided, especially regarding Ecuador's urbanization trends and the impact of neoliberal policies, adds depth to the analysis. The manuscript addresses the gap in existing literature by concentrating on intermediate Andean cities, such as Loja, which have been understudied despite being significant contributors to contemporary urban growth.

The results section of the manuscript sheds light on the consolidation process and the fragility of the San Cayetano neighborhood, offering valuable insights into the dynamics of its spatial transformation. The scientific novelty lies in the meticulous analysis of the territory, providing quantitative data on the consolidation process, land use patterns, and architectural typologies. The study's merit is evident in its identification of San Cayetano as a third space—a diaphragm integrating urban and rural practices, challenging conventional interpretations of rural-urban interfaces.

The consolidation process, exemplified by the expansion of the neighborhood within the urban boundary, is well-documented, showcasing a 65.29% increase in territory between 2009 and 2019. However, the manuscript would benefit from a more detailed explanation of the factors influencing this process. The prevalence of agricultural activities and vacant lots due to geological constraints is mentioned, but a deeper exploration of these constraints and their impact on consolidation would enhance the reader's understanding.

The examination of the territory's fragility and the identification of three block typologies contribute to the novelty of the study. The recognition of a peri-urban edge characterized by heterogeneity and fragmentation is a valuable contribution to the understanding of rururban environments. The inclusion of figures depicting the urban fabric and block typologies enhances the clarity of the presentation.

To improve the manuscript, further elucidation on the architectural forms and the impact of topography on the road system would add depth to the analysis. Additionally, the study mentions the ongoing growth and urbanization of San Cayetano but could provide more clarity on the implications of this growth for the residents and the broader urban context. Addressing these aspects would strengthen the overall scientific contribution of the study.

The discussion section of the manuscript delves into the implications of the rapid transformation from rural to urban areas, emphasizing the impact on the sense of identity and belonging within the San Cayetano neighborhood. The identification of a collective urban image in the making and the influx of low and middle-income groups into peri-urban areas for personal housing development are highlighted. The discussion effectively connects societal criteria, topography, and the urban form of the neighborhood, illustrating how challenges in the landscape contribute to irregular and multifarious plots.

The identified mono-functional and low-density usage patterns are thoughtfully discussed, revealing spatial and social segregation, as well as a continued dependence on services from the city center. The study's hypothesis of a spatial occupation that defies strict categorization is well-supported, acknowledging the intertwining of urban and rural characteristics within San Cayetano. However, the discussion could benefit from a more nuanced exploration of the specific vulnerabilities generated by informal construction practices, inadequate infrastructure, and the pervasive sense of insecurity among inhabitants.

In the conclusion section, the manuscript appropriately summarizes the commodification of land in the wake of liberalization and the subdivision of rural areas to maximize profitability. The acknowledgment of the third space, blending rural and urban elements, is reiterated, emphasizing the diversity of landscapes and productive practices. The discussion on challenges faced by these cities, including inadequate urban systems, environmental degradation, and limited access to services, is a valuable addition.

To enhance the paper further:

1.     Clarify the specific impacts of the influx of low and middle-income groups on the sense of belonging and the emerging collective urban image within San Cayetano.

2.     Provide more detailed insights into the vulnerabilities generated by informal construction practices, lack of risk assessments, and inadequate provision of physical infrastructure, and explore potential strategies for addressing these issues.

3.     Propose specific recommendations or interventions for sustainable urban development and improving living conditions, taking into account the challenges identified in the discussion and conclusion sections.

Author Response

(The authors gave the same response as above.)

Reviewer 4 Report

Comments and Suggestions for Authors

The article, in my opinion, unfortunately lacks proper preparation. Regarding the introduction and literature review (the latter of which is almost nonexistent), there is a notable absence of fundamental definitions and criteria for “urban area” and “rural area.” What do these categories truly encompass? How do researchers from different parts of the world define them? Furthermore, what exactly is meant by the term “third space”? For some, it may refer to a place we pass by without paying attention—a vacant space that people do not consider their own. For others, it signifies a social space distinct from home and work, an informal public place where people gather. There are indeed numerous interpretations and applications.

Without the above, it is difficult to imagine the correct execution of the study—especially since the method description lacks information on which factors are considered and to what extent when determining what qualifies as an urban versus rural area. I understand that what remains should be labeled as ‘third spaces’ according to the authors; however, from my perspective, these areas align more with transitional zones (akin to the ‘shadow zone’) —a concept previously identified (albeit somewhat differently than today) in Ernest Burgess’s 1925 model. Additionally, clear criteria for classifying basic objects are missing. Perhaps a hexagonal grid or one of the fundamental scoring methods could be employed. Alternatively, grouping similar objects using either pattern-based or non-pattern-based methods could yield valuable insights, especially when tested through field research.

Another concern is the data selection and the presentation of figures. What are the parameters of the existing maps used in the study? What is their source? Is there any data error? Are there differences in the projection system (given that the maps may be from different periods), and which software was used for the analysis? Furthermore, the figures should be translated into English before printing.

Lastly, but crucially, there is a lack of discussion. How do the presented results align with other research? What has been confirmed, and what has been challenged? What is unique about the construction of the example presented? How might this impact the application of the presented method in other parts of the continent (or on other continents)?

Author Response

(The authors gave the same response as above.)

Reviewer 5 Report

Comments and Suggestions for Authors

The paper deals with a fascinating case. We rarely get information from Ecuador, especially from the remote medium-sized towns. Therefore, the work is precious and will be an essential reference for professionals working on the development of medium-sized cities. I recommend its publication with minor changes.

It would be important to explain several elements that contributed to the city's development and are unknown to outsiders, such as the banana crisis, the taking of the graveyard land, the dollarization, and the economic crisis.

In the same way, the administrative relationship between the city and the neighbourhood under study should be described; what is the municipal system like in Ecuador, e.g. what is the role of the parish? Self-governments? There should be a map of where the municipality is located in the country.

It would be important to improve the maps and figures. Each one should have a title, including the photos. In many places, there is no explanation of the symbology (legend): Figur 3 – missing legend: Does the frame represent the neighbourhood under study?

Elsewhere, there are also legends and explanations in Spanish (figs. 4, 5, 8).

The literature is based on Spanish-language literature. It would be good to refer to works written in English about the region or country, as well as international works with which to compare local conditions.

I am very interested in the work, so I would happily contact the authors.

Author Response

(The authors gave the same response as above.)

Reviewer 6 Report

Comments and Suggestions for Authors

First of all, I would like to thank the author(s) for their work and the opportunity to review this paper. On the positive side, I would like to point out that it is an easy-to-read, engaging, and entertaining piece for the reader. It presents an analysis of a peri-urban area in the Andean region and its urban evolution towards integration with the consolidated city. The visual contribution through images and maps is also interesting.

However, there are some aspects that, in my opinion, detract from the work. Although the topic may initially be interesting by characterizing a good case study, I miss a deeper analytical approach. The work is purely descriptive, and although it is noted at the beginning that the economic analysis is beyond the scope of the article's purpose, it would be highly recommended to analyze, even superficially, aspects such as the socio-demographic or labor composition of the area under analysis.

Ultimately, the work is too superficial and descriptive, and in my opinion, it would be necessary to delve deeper into the analysis (either qualitatively or by expanding the analysis towards a more quantitative study).

The issue of mixed uses of rural-urban space could have more potential from my point of view. Additionally, the use of certain concepts requires a broader explanation, such as the use of the concept of neoliberal city, the relationship between neoliberal policies and the emergence of peripheries beyond capital concentration, the relationship between the three historical moments (banana exploitation crisis, agrarian reform, and so-called substitution processes) in the emergence of informal settlements.

The intensification of tertiary activities in urban centers is a globally expanding phenomenon that, however, has not necessarily led to the proliferation of informal settlements in urban peripheries. Why in the Andean region and not in other regions?

Finally, I would like to mention some very minor considerations. A detailed review of the text is necessary as I have noticed some typos (for example: in the second paragraph of the section 2 (materials and methods) or in the Note of figure 4). Some years in the bibliography section are in bold and others are not. It is also important to mention Figure 8, as the text is in Spanish, requiring its translation into English.

Author Response

(The authors gave the same response as above.)

Round 2

Reviewer 1 Report

Comments and Suggestions for Authors

ok

Author Response

Thank you very much for your suggestions for the document.

Reviewer 2 Report

Comments and Suggestions for Authors

The manuscript has been improved. However, there are still some points to be improved and supplemented.

Agree. We have given the focus to the fractions that are produced by the topography of the site producing fragments that affect the natural and built landscape, which is reflected in the different maps presented. The third space, as you indicate, has a broader meaning, so we believe that the focus on fractions is the right one”: That was not the point of view that I raised in the previous review. Since you use a term (“Third space”) that a globally known urban researcher has already done before you, and many people have referred to, you need to go into more detail about how the concept you use differs from his one. However, this still did not happen in the manuscript, and Soya's name is not mentioned. And from this point of view, I assume that it does not matter how you interpret this term, because in the case of internationally known and widely cited researchers, you are expected to clarify the difference if they come up with a concept sooner than you do.

The manuscript contains relatively few "research limitation" sections, although there were obviously some due to the nature of your choice of topic and the research methods. These should be added, for example at the end of the study.

For all illustrations, it would be worthwhile, as is customary in scientific studies, to indicate the year of edition (which is not uniform, only indicated in some places in your manuscript) and authors (even if they are you). And in order for the global reader to be able to relate spatially, it is necessary to use scale bars on all maps (including the right-hand part of the first figure, Figure 2). The map used in Figure 9 has a low resolution and therefore does not show the numbering well.

Comments on the Quality of English Language

The paper reads relatively well. 

Author Response

In the document, what has been enlarged has been marked in gray.

Reviewer 4 Report

Comments and Suggestions for Authors

Thank you for making corrections and additions. I believe the text has benefited from them. Nevertheless, I have one more general comment. The discussion, as the name suggests, is a place to relate the study’s findings to the broader context found in the literature. Therefore, I reiterate and uphold the points from the fourth comment, particularly: How do the presented results align with other research? What has been confirmed, and what has been challenged? In my opinion, this section cannot be written without referencing other authors. Additionally, please explicitly state which GIS tools were used. Different programs are designed for various tasks, and the operations and procedures applied in them may differ (based on slightly different methodologies), which could impact the obtained results. Overall, I consider the direction of changes in the text to be positive and eagerly await the next version.

Author Response

(The authors gave the same response as above.)

Reviewer 6 Report

Comments and Suggestions for Authors

Many thanks to the authors for their effort and for allowing me to review this second version of the work, which is notably improved. In my opinion, the suggested aspects have been correctly incorporated and the work has gained strength and understanding.

Author Response

In the document, what has been enlarged has been marked in gray. Please see the attachment.
